# Population Structure and Association Mapping for Agronomical and Biochemical Traits of a Large Spanish Apple Germplasm

**DOI:** 10.3390/plants12061249

**Published:** 2023-03-09

**Authors:** Pierre Mignard, Carolina Font i Forcada, Rosa Giménez, María Ángeles Moreno

**Affiliations:** Department of Pomology, Estación Experimental de Aula Dei, Consejo Superior de Investigaciones Científicas (EEAD-CSIC), P.O. Box 13034, 50080 Zaragoza, Spain

**Keywords:** antioxidants, genetic association, inter-chromosomal linkage disequilibrium, *Malus × domestica* borkh, simple sequence repeats-SSRs, SWEETs genes

## Abstract

A basic knowledge of linkage disequilibrium and population structure is necessary in order to determine the genetic control and identify significant associations with agronomical and phytochemical compounds in apple (*Malus × domestica* Borkh). In this study, 186 apple accessions (Pop1), representing both Spanish native accessions (94) and non-Spanish cultivars (92) from the EEAD-CSIC apple core collection, were assessed using 23 SSRs markers. Four populations were considered: Pop1, Pop2, Pop3, and Pop4. The initial Pop1 was divided into 150 diploid (Pop2) and 36 triploid accessions (Pop3), while for the inter-chromosomal linkage disequilibrium and the association mapping analysis, 118 phenotype diploid accessions were considered Pop4. Thus, the average number of alleles per locus and observed heterozygosity for the overall sample set (Pop1) were 15.65 and 0.75, respectively. The population structure analysis identified two subpopulations in the diploid accessions (Pop2 and Pop4) and four in the triploids (Pop3). Regarding the Pop4, the population structure with K = 2 subpopulations segregation was in agreement with the UPGMA cluster analysis according to the genetic pairwise distances. Moreover, the accessions seemed to be segregated by their origin (Spanish/non-Spanish) in the clustering analysis. One of the two subpopulations encountered was quite-exclusively formed by non-Spanish accessions (30 out of 33). Furthermore, agronomical and basic fruit quality parameters, antioxidant traits, individual sugars, and organic acids were assessed for the association mapping analysis. A high level of biodiversity was exhibited in the phenotypic characterization of Pop4, and a total of 126 significant associations were found between the 23 SSR markers and the 21 phenotypic traits evaluated. This study also identified many new marker-locus trait associations for the first time, such as in the antioxidant traits or in sugars and organic acids, which may be useful for predictions and for a better understanding of the apple genome.

## 1. Introduction

The modern cultivated apple (*Malus × domestica* Borkh) is the most significant and ancient fruit crop of the *Rosaceae* in the world as well as in Spain [1]. In 2021, more than 93 M tonnes of apples were produced around the world [2]. In 2021, an average of 617 thousand tonnes was produced in Spain [3]. However, more than 516 thousand tonnes of the national production were dessert apples for fresh consumption and juices, while cider apple production reached only 100 thousand tonnes. Nevertheless, apple production around the world is based on a reduced number of modern bred varieties [1,4,5]. In fact, apple production is dominated by some well-adapted modern cultivars (‘Fuji’, ‘Gala’, ‘Golden’, ‘Granny Smith’, and ‘Delicious’), many of which are genetically linked [6,7]. In addition, the ‘Golden Delicious’ cultivar represented more than 46% of the Spanish apple production for fresh consumption with more than 240 thousand tonnes [3]. As a consequence, many of the traditional and/or locally well-adapted cultivars have been considered outdated, and a dramatic loss of genetic diversity has been noticed in the apple gene pool in the last decades [8,9,10,11,12,13].

All commercial apple cultivars have a basic chromosome number of 17 and are considered to have evolved after an autopolyploidization of a *Gillenia*-like taxon followed by diploidization [14,15]. Most domesticated apple cultivars are diploid (2n = 2x = 34), even though triploid (2n = 3x = 51) and tetraploid (2n = 4x = 68) cultivars can also be found. The modern apple should be the result of a long evolutionary process spanning hundreds of years, with several species contributing to the domesticated apple gene pool encountered nowadays [16].

Despite the high genetic variability of the apple gene pool, breeding programs over the years have been mainly based on a few organoleptic traits, aesthetic criteria, and resistance to specific diseases [10], resulting in a small number of cultivars and a loss in diversity. Moreover, the breeding programs have limited and reduced the use of the total number of cultivars, such as ‘Cox’s Orange Pippin’, ‘Golden Delicious’, ‘Jonathan’, ‘Red Delicious’, and ‘McIntosh’, although more than 10,000 different apple accessions have been described globally [6,10,17]. Indeed, a great number of new cultivars were obtained from breeding programs during the twentieth century from a reduced number of progenitors and, consequently, shared a high degree of parentage [6,17,18]. For this reason, germplasm evaluation, characterization, and conservation of genetic diversity are crucial for the management of genetic properties [19]. Therefore, there is an increased need for studying and collecting old local apple accessions that could provide a better knowledge of the history of the crop and could be used in future breeding programs aiming to select better adapted cultivars to climate conditions in the current context of global climate change [5,19,20]. In fact, apple allelic diversity should be used to address existing and future biotic and abiotic issues with respect to production [8]. Furthermore, in the last few decades, interest in the genetic and molecular characterization of apples has increased. The diversity found at the genetic level between apple accessions reflects a combination of historical selections and adaptive processes, resulting in extensive genetic variation but a limited population structure [6].

The microsatellite markers or Simple Sequence Repeats (SSRs) have been in apple and other fruit crops, in general, one of the most used markers for molecular characterization together with the Single Nucleotide Polymorphism markers (SNPs). The genome sequence of the domesticated apple has been released and accounts for approximately 750 Mb per haploid [15]. The SSRs analysis is a good technique within crop species, due to their abundance, codominance, high polymorphism, easy use with the PCR conditions, and relatively low cost [21,22]. Indeed, many studies based on these molecular markers have assessed the diversity of *Malus* accessions [1,8,9,11,14,23,24]. Gross et al. [14] demonstrated that nine SSR loci were sufficient to determine potential duplicate and study differences among *Malus* accessions. In the present work, 23 microsatellites markers were used for the molecular characterization of a large set of accessions from the northeastern part of Spain. The European Cooperative Program for Plant Genetic Resources (ECPGR) has published several lists of recommended markers, including SSRs that span most of the apple genome [8,25] and have been tested on a set of standard *Malus* accessions. Many of these mapped SSRs have been associated with quantitative trait loci (QTL) with a great interest in agronomical, morphological, and/or organoleptic traits and could be used as molecular tools for marker assisted selection (MAS) in future plant breeding programs [10,26].

Apart from the recommended SSR, the present work aims to report the identification of the MdSWEET genes by SSRs markers designed by Zhen et al. [27]. In plants, SWEET transporters function as bidirectional uniporters that mediate the influx and efflux of sugars across cell membranes. SWEETs can be divided into four subgroups [27,28]. Clades I, II, and IV of the SWEETs transport predominantly hexoses, whereas clade III of the SWEETs appears to be sucrose transporters (SUTs) [27]. This study will allow a better understanding of the effect of SWEETs on fruit sugar accumulation, and it will also be helpful for the genetic improvement of fruit sugar accumulation in apple-breeding programs.

Moreover, association mapping, also known as linkage disequilibrium (LD) mapping, relies on the strength of association between genetic markers and phenotype. Therefore, LD mapping is an approach that detects and locates genes relative to an existing map of genetic markers [29,30]. Consequently, this method detects relationships between phenotypic variation and gene polymorphism in existing germplasm and in unrelated individuals. Association mapping has been successfully used to identify genes involved in flowering and ripening traits in apples [12], although the bibliography for organoleptic and fruit quality traits is scarce [31,32,33,34,35].

The present work aims to study the genetic characterization of the *Malus × domestica* Borkh germplasm located at the Experimental Station of Aula Dei (EEAD-CSIC), Spain, to assess the population structure, inter-chromosomal linkage disequilibrium, and association mapping between the local well-adapted accessions compared with the modern and commercial cultivars. This work provides, thus, molecular tools for genetic improvement of fruit quality in apple breeding programs as well as a better knowledge of the apple genetic resources available through a common set of 23 SSR markers. Our results will highlight the importance of genetic variation in germplasm collections for the effective conservation of biodiversity and phytochemical resources in the domesticated apple crop.

## 2. Materials and Methods

### 2.1. Plant Material and Field Trial

This research counted 186 accessions (*Malus × domestica* Borkh) from the apple germplasm bank (Figure 1; Table 1) established at the Experimental Station of Aula Dei (EEAD-CSIC, Zaragoza, NE Spain: 41°43′42.7″ N, 0°48′44.1″ W). The 186 genotypes (Pop1 for population number 1) were grown under Mediterranean soil conditions, typical of the Central Ebro Valley area. This geographic area is characterized by a semi-arid climate with warm and dry summers, high radiation, and large day–night temperature variation [18]. The accessions assessed in this work (Table 1) consisted of a wide range of geographic origins (94 Spanish and 92 non-Spanish accessions). Indeed, most of the non-Spanish accessions were commercial cultivars, whereas autochthonous commercial cultivars or traditional landraces represented the local-Spanish accessions. The accessions were categorized according to Mignard et al. [5,20]. In the field, each accession had three-tree replications established in a single block design. Trees were trained to a low-density system (6 m × 5 m), and the cultural management was carried out as in a commercial plantation. The orchard was flood irrigated every 12 days during the summer.

### 2.2. Leaf and Fruit Sampling

For the evaluation of the ploidy level for the 186 accessions (Pop1) of this work, newly expanded leaves were collected from each accession and analyzed as described in Reig et al. [18] by flow cytometry. The accessions were consequently classified into diploids (Pop2: 150 accessions) and triploids (Pop3: 36 accessions). According to the fruit sampling, 30 fruits (10 fruits × tree × rep.) were harvested when fruit firmness (FF) attained a value around 70–80 N or when they displayed the representative peel color of each accession. Accessions were harvested during at least three seasons within the 2014–2018 period, and means for each year and accession were calculated [20]. The maturity date fluctuated from late June to early December.

### 2.3. Phenotypical Evaluation of Biochemical Traits

#### 2.3.1. Basic Fruit Quality

Soluble solids content (SSC) and titratable acidity (TA) were determined on flesh juice as described by Mignard et al. [20]. SSC results were expressed as °Brix, and TA results were expressed as g of malic acid per liter. The ripening index (RI) was thus calculated based on the SSC/TA ratio.

#### 2.3.2. Antioxidant Compounds, Vitamin C, and Relative Antioxidant Capacity

For the analysis of total phenolic content (TPC), total flavonoid content (TFC), vitamin C (ascorbic acid–AsA), and the relative antioxidant capacity (RAC), a flesh sample composite of 5 g of five peeled fruits per replicate was frozen in liquid nitrogen and kept at −20 °C until further analysis. The biochemical compounds were analyzed using a 96-well microplate as described by Font i Forcada et al. [36]. 

TPC was determined using the Folin-Ciocalteau method [37] with modifications, and the results were expressed in mg of gallic acid equivalent (GAE) per 100 g of fresh weight (FW). TFC was determined using a colorimetric assay based on the method of Zhishen et al. [38] with minor modifications, and the results were expressed in mg catechin equivalent (CE) per 100 g FW. The RAC was measured using the 2,2-diphenyl-1-picrylhydrazyl (DPPH) method adapted from Brand–Williams et al. [39] with modifications. The results were expressed in mg of Trolox per 100 g FW. Finally, vitamin C–ascorbic acid (AsA) was determined using the method for the spectrophotometric determination of AsA as described by Zaharieva and Abadía [40] with modifications. The results were expressed in mg AsA per 100 g FW.

#### 2.3.3. Individual Sugars and Organic Acids

Individual sugars and organic acid contents were assessed by HPLC, as reported by Mignard et al. [5]. Sugars were analyzed using an Aminex HPX-87C column (300 mm × 7.8 mm, Biorad, Hercules, CA, USA) with a refractive index detector at 35 °C (Waters 2410, Waters Corporation, Milford, MA, USA) and milliQ water at 85 °C as mobile phase. Organic acids were assessed with a Rezex™ ROA-Organic Acid H+ (8%) column (300 mm × 7.8 mm, Phenomenex) with a photodiode array detector (Waters 2489, Waters Corporation, Milford, USA) at 210 nm and a sulphuric acid solution (0.005 N) at room temperature as mobile phase. Individual sugars (glucose, fructose, and sucrose), the sugar-alcohol (sorbitol), and main organic acids (citric, malic, oxalic, quinic, succinic, shikimic, and tartaric acids) concentrations were determined by their characteristic retention times following standards and expressed as g per kg of FW.

### 2.4. Microsatellite Loci Analysis and Genotyping

For genomic DNA extraction, young leaves were collected from each accession, frozen instantly in liquid nitrogen, and stored at −20 °C until use. DNA was isolated using the NucleoSpin^®^ Plant II kit (Macherey-Nagel, Düren, Germany) according to the manufacturer’s instructions. DNA quality was examined on agarose gels (1%), and DNA concentration was determined by spectrophotometry using a NanoDrop^™^ (Waltham, MA, USA). Twenty-six simple sequence repeat (SSRs) markers previously described in *Malus* were tested in the apple population (Table 2), but three of them exhibited weak or no amplification in our plant material. Eleven of these 23 SSRs, which amplified, were recommended by the ‘European Cooperative Program for Plant Genetic Resources (ECPGR) *Malus*/*Pyrus* working group’ with a standard methodology proposed to allow comparisons of the same accessions between different laboratories [41]. In addition to these eleven SSRs, four SSRs have also been assessed in apple studies [6], and other eight SSRs were specially designed to amplify in areas of the genome in relation to the metabolic pathway of sugars: the ‘Sugar Will Eventually is Exported Transporters–SWEET’ genes [27].

Six different multiplexed reactions were used for these 23 SSRs. Forward SSR primers were labeled with 5′-fluorescence dyes, including PET, NED, VIC, and 6-FAM. The polymerase chain reactions (PCR) for the multiplexed PCRs were performed in a final volume of 10 μL using 10 ng of DNA template, 0.1 μM of each primer (with the exception of some markers as described in Table 2), and 1× PCR Master Mix of the QIAGEN kit multiplex PCR (Qiagen, Hilden, Germany). Two PCR cycling conditions were used. The first PCR cycling was as follows: pre-incubation for 15 min at 95 °C, followed by 5 cycles using a touchdown amplification program with an annealing temperature reduced by 1 °C per cycle from 65 °C to 60 °C. The next step involved 30 cycles, each consisting of 30 s denaturing at 95 °C, 60 s annealing at 60 °C, and 60 s elongation at 72 °C. The last cycle ended with a final 30 min extension at 72 °C. The second PCR cycling consisted of a 15-min pre-incubation at 95 °C, followed by 7 cycles with a touchdown amplification program from 65 °C to 58 °C, followed by 30 cycles, each consisting of 30 s denaturing at 95 °C, 60 s annealing at 58 °C, and 60 s elongation at 72 °C. The last cycle ended with a final 30 min extension at 72 °C. Fragment analysis and sizing were carried out using Geneious Prime v.2022.0.1 software (Thermo Fisher Scientific, Waltham, MA, USA). The PCR products were diluted with milliQ water at 1:5 (*v*/*v*) and mixed with 12 μL Hi-di Formamide (Applied Biosystems) and 0.5 μL size standard Gene Scan^™^ 600 Liz^®^ (Applied Biosystems). Finally, the fragment analyses were performed using an Applied Biosystems 3130 DNA Genetic Analyzer (Applied Biosystems, Foster City, CA, USA). 

### 2.5. Data Analysis for the Whole Dataset

#### 2.5.1. Phenotype Statistical Analyses

All statistical analyses were carried out using the R language [42]. A one-way analysis of variance (ANOVA) with a *p* ≤ 0.05 was run to determine whether there were any statistically significant differences between the means of the evaluated traits. Moreover, Pearson’s bivariate correlations were performed to better determine how biochemical traits contribute to variability among accessions.

#### 2.5.2. Diversity and Variability Assessment

Genetic parameters were carried out with the 23 microsatellites and for the whole dataset (Pop1), which corresponds to the 186 accessions from the EEAD–CSIC germplasm bank (186 accessions in total, divided into two different pools: 150 diploids, Pop2; and 36 triploids, Pop3). No multiloci SSR marker was detected in this study. The number of observed alleles per locus (N_A_), the effective number of alleles per locus (N_E_) [43], and rare alleles (N_B_: alleles with a frequency below 5%) were determined using the Genodive software [44]. Observed heterozygosity (H_o_: number of heterozygous accessions/number of accessions assessed), expected heterozygosity (H_e_ = 1 − ∑*ρ_i_*^2^, where *ρ_i_* is the frequency of the i^th^ allele) [45], Wright’s fixation index (F_is_ = 1 − H_o_/H_e_), and Shannon’s information index (I) [46] were calculated using the PopGene 1.32 software [47] (http://www.ualberta.ca, accessed on 15 December of 2022).

#### 2.5.3. Analysis of Population Structure

Population structure analysis was first performed on the whole dataset (Pop1). All the accessions were categorized as Spanish or non-Spanish accessions. The program STRUCTURE (version 2.3.4) implements a model-based clustering criterion for inferring population structure using genotypic data from unlinked markers [48]. All kinds of models, including both “ancestry” and “allele frequency” models, were fitted with the selection of admixture and allele frequency correlation, respectively. Furthermore, we also performed ten independent runs per K value with a 10,000 burn-in period and 100,000 MCMC replications, starting with K = 1 to K = 6, under the admixture model. The statistic ΔK was then carried out, where K specifies the number of subpopulations or clusters. This analysis was based on the rate of change in the log probability of the data [49] to select the optimum number of K subpopulations for each population assessed. 

### 2.6. Data Analysis for the Association Mapping for 118 Accessions

Genetic parameters for trait marker association were carried out for the 23 SSRs and according to the Pop4, corresponding to 118 phenotype diploids. Indeed, from the Pop2 (150 diploids), only 126 accessions were phenotyped, and eight resulted in duplicates (126 − 8 = 118 diploids: Pop4). In fact, from these 126 accessions and using the software Cervus v.3.0.7 [50], genetic uniqueness and redundancy were eliminated. The multi-locus DNA profile of all the accessions was compared pairwise under the identity analysis, setting the minimum number of matching loci at 23 and zero mismatches. Eight accessions resulted as duplicated diploid accessions from pairwise comparison of SSR profiles analyzed by Cervus software. The ‘Averdal-1’, ‘Averdal-2’, ‘Evasni’, ‘Red Elstar’, ‘Royal Red Delicious’, ‘Starkrimson-2’, and ‘Topred Delicious’ cultivars were duplicated among them, and thus, only one accession was kept in the study, the ‘Averdal-1’. Moreover, the ‘Galaxy’ and the ‘Regal Prince-1’ were duplicated between them, and the ‘Verde Doncella-1’ and ‘Verde Doncella-2’ were also duplicated between them. The ‘Galaxy’ and ‘Verde Doncella-1’ accessions were kept in the study. 

#### 2.6.1. Inter-Chromosomal Linkage Disequilibrium

The analysis of inter-chromosomal linkage disequilibrium (LD) was calculated using the trait analysis software by Association, Evolution, and Linkage (TASSEL, version 3.0.174, http://www.maizegenetics.net, accessed on 21 November of 2022). Alleles with frequencies below 5% were removed (minor allele frequency-MAF). Inter-chromosomal linkage Disequilibrium between pairs of multiallelic loci was calculated using the r^2^ coefficient, separately for loci in the same or different linkage groups (LG). The statistical r^2^ gives an indication of both recombination and mutation [51]. The significance level of LD between loci was examined using a permutation test implemented in TASSEL software for multiallelic loci, using the ‘rapid permutation’ option.

#### 2.6.2. Association Mapping

TASSEL (v.3.0.174) was used with the General Linear Model (GLM) option [52] to examine associations between the phenotypic traits and the 23 SSR DNA markers. A structured association approach could be corrected for false associations using a Q-matrix of population membership estimates [52]. Therefore, the population membership estimates obtained from STRUCTURE analyses were fitted as a covariate in the GLM, where phenotype = population structure + marker effect + residual. As permutation methods can provide exact control of false positives and allow the use of non-standard statistics that make only weak assumptions about the data, a standard correction for multiple testing consisting of 10,000 permutations was run before carrying out the GLM.

## 3. Results

### 3.1. Phenotypic Evaluation and Pearson’s Correlations

The phenotypic evaluation and the statistical analysis were carried out for the Pop4 (118 diploids) out of the 186 apple accessions described in Table 1 and more extensively by Mignard et al. [20]. The one-way ANOVA analysis showed significant differences (*p* ≤ 0.001) among the different apple accessions for all traits evaluated (Table 3). 

The SSC ranged among apple accessions from 10.14 (‘Bellaguarda Lardero’) to 17.03 (‘Eugenia’) °Brix. Regarding the TA, values varied greatly, ranging from 1.77 (‘Verde Doncella’) to 17.29 (‘Reguard-2’) g of malic acid per liter. The RI values ranged from 0.76 (‘Reguard-2’) to 8.55 (‘Verde Doncella’). The standard deviations for SSC, TA, and RI were fitted at 1.36, 2.92, and 1.34, respectively (Table 3). The TPC varied greatly among accessions, ranging from 15.24 (‘Biscarri-1’) to 98.07 (‘Camuesa Fina de Aragón’) mg GAE/100 g FW. For the TFC, values ranged from 6.00 (‘Biscarri-1’) to 88.95 (‘Camuesa Fina de Aragón’) mg CE/100 g FW. Regarding AsA, values ranged from 1.37 (‘Delgared Infel’) to 5.30 (‘Transparente’) mg AsA/100 g FW. Finally, the RAC values ranged from 5.93 (‘Delgared Infel’) to 30.82 (‘Les-1’) mg trolox/100 g FW. The standard deviations for TPC, TFC, AsA, and RAC, respectively, were fitted at 15.76, 14.67, 0.82, and 5.08 (Table 3).

Total sugar values (sugars) ranged significantly among apple accessions and years from 63.69 (‘Transparente Blanca’) to 115.27 (‘Fuji’) g kg^−1^ FW. Regarding the main individual sugars, fructose values ranged from 31.39 (‘Baujade’) to 61.41 (‘Akane’) g kg^−1^ FW, and the alcohol sugar sorbitol values varied from 1.20 (‘Plaona’) to 11.96 (‘Prau Riu-3’) g kg^−1^ FW. The standard deviations for total sugars, fructose, and sorbitol, were 9.87, 5.17, and 2.43, respectively. Total acid values (Acids) ranged among accessions and years from 3.42 (‘Verde Doncella’) to 11.39 (‘Astrakan Red’) g kg^−1^ FW. Regarding the main individual acids, malic acid ranged considerably compared with the others, from 2.68 (‘Delciri’) to 9.81 (‘Reguard-2’) g kg^−1^ FW. The standard deviations for total acids and malate were fitted at 1.82 and 1.69 g kg^−1^ FW, respectively.

Significant (*p* ≤ 0.01) bivariate correlations were found between the different traits evaluated except for the peel color and glucose which did not show any correlation (Figure 2). Positive and significant (*p* ≤ 0.01) correlations were found between phenolics and flavonoids (r = 0.96) and between phenolics, flavonoids, and RAC (r = 0.90 and r = 0.87, respectively). Significant positive correlations were also found between the antioxidant compounds and the organic acids. The individual sugars and the sum of sugars showed significant but low negative correlations with the bioactive compounds (TPC: r = −0.21, TFC: r = −0.26, and RAC: r = −0.25) and the organic acids. Moreover, high and significant correlations were found between harvest date and concentrations of soluble solids, sorbitol, sugars, and, as expected, with the ripening index. Significant and negative correlations were also found between the harvest date and individual/total organic acids, the titratable acidity, the antioxidant compounds (TPC and TFC), and the relative antioxidant capacity (RAC) (Figure 2). According to Pearson’s correlations, the non-Spanish accessions seemed to present lower organic acids and antioxidant compounds (TPC, TFC, and RAC) concentrations compared with the Spanish accessions.

### 3.2. Genetic Diversity

The mean estimated values for genetic parameters based on the 23 SSR loci assessed are presented in Table 4. According to Pop1 (186 accessions), the 23 microsatellites were all polymorphic, and amplicons could be observed for all of them, for a total of 360 alleles. The average number of alleles per locus (N_A_) was 15.65, ranging from 4 (MdSWEET2b) to 35 (CH01f02). Nevertheless, the number of effective alleles per locus (N_E_) was significantly lower (Table 4). Indeed, the average number of effective alleles (N_E_) overall the loci was 4.90, ranging from 1.5 (MdSWEET2e) to 8.98 (CH02d08). Moreover, rare alleles (N_B_) were found in all loci, and the number of them increased with the number of alleles per locus (r = 0.96). Alleles with a frequency lower than 5% (N_B_) varied from 33.3% at locus CH02c11 (5 out of 15) to 86.7% at locus CH04e05 (26 out of 30). Furthermore, all loci except for five (CH04e05, CH01f02, CH01h01, MdSWEET2e, and MdSWEET2d) were not in Hardy-Weinberg equilibrium (*p* ≤ 0.05). In fact, the 186 accessions assessed in this study do not belong to a panmictic population. While mean observed heterozygosity (H_o_) was 0.75, ranging from 0.19 (MdSWEET2e) to 0.96 (MdSWEET7b) (Table 4), average expected heterozygosity (H_e_) was 0.77, varying from 0.41 (MdSWEET2e) to 0.92 (CH05f06). F_is_ values were positive in 10 primers and negative in the remaining 13 SSRs, indicating a high level of heterozygosis in the genotypes assessed.

Among the Pop2 (150 diploid accessions) and Pop3 (36 triploid accessions), the average number of alleles per locus (N_A_) was 14.7 and 10, respectively, and the number of effective alleles per locus (N_E_) was fitted at 5.23 (Pop2) and 4.66 (Pop3). The observed and expected heterozygosities were higher in diploids (H_o_ = 0.79; H_e_ = 0.77) than in triploids (H_o_ = 0.71; H_e_ = 0.73). The F_is_ value was negative for the Pop2, while it was positive for the Pop3.

Pairwise comparison of multiloci profiles revealed eight duplicated diploid accessions with redundancies, leading to the removal of redundant accessions before further analyses (4.3% of redundancy). Several cases of homonymy (i.e., accessions with the same name but different profiles according to the 23 SSRs) were also found (‘Camosa-1’ and ‘Camosa-2’, ‘Jonathan-1’ and ‘Jonathan-2’, ‘Granny Smith-1’ and ‘Granny Smith-2’, ‘Mañaga-1’ and ‘Mañaga-2’, ‘Starking-1’ and ‘Starking-2’, and ‘Verde Doncella-1’ and ‘Verde Doncella-3’). The final number of unique diploid genotypes further analyzed was therefore 118 (Pop4).

### 3.3. Population Structure

Bar plots were obtained according to the values of K, the assumed number of subpopulations, corresponding to the number of clusters defined by ΔK [49] (Figure 3). Model-based clustering analyses were used to determine the genetic diversity structure within the 186 (Pop1) assessed *Malus × domestica* (Borkh) genotypes. A pairwise STRUCTURE analysis based on 23 allelic SSRs molecular data was carried out, and the maximum value of ΔK was observed at K = 2 for Pop2 (150 diploids) and Pop4 (118 diploids), suggesting two genetic clusters. Along with the 36 triploids in the study (Pop3), the maximum value of ΔK was revealed at K = 4, although at K = 2, an increase in ΔK was observed (Figure 3). Genotypes were divided into two or four clusters, respectively, based on their membership coefficients (Q), considering the genotypes as pure when the membership coefficient (qI) was greater than 0.80 and as an admixture or hybrid when the membership coefficient (qI) was lower than 0.80 [48].

The Pop2 (150 diploids) was represented by 72 Spanish and 78 non-Spanish accessions, the Pop3 (36 triploids) by 22 Spanish and 14 non-Spanish, and the Pop4 (118 diploids) by 72 Spanish and 46 non-Spanish accessions. However, the separation at K = 2 was congruent for all the diploid populations assessed, and admixture in each subpopulation was observed, demonstrating allele sharing (Figure 3). For Pop4 (118 diploids), the mean values of genetic differentiation (F_st_) among the two subpopulations obtained with K = 2 were 0.25 and 0.0024, respectively. The F_st_ of the first subpopulation indicated a strong genetic differentiation for the accessions of this group. In addition, the allele frequency divergence among subpopulations was 0.0678 and demonstrated significant genetic differences between subpopulations (>0.05). Furthermore, regardless of the populations studied (Pop2 and Pop4), the first subpopulation was represented mostly by Spanish accessions and the second by the non-Spanish reference cultivars.

Additionally, the dendrogram obtained by the unweighted pair group method with arithmetic mean (UPGMA) analysis was compared with the phenotypic data (Figure 4) and the two subpopulations of the STRUCTURE results for the Pop4 (Figure 5). In the dendrogram obtained from the similarity matrix of pairwise analyses for the 23 SSR markers, there is a clear agreement between clusters representing genetic diversity and subpopulations obtained with STRUCTURE at K = 2. Furthermore, the population assessed can clearly be divided into six clusters/groups from the first two major clusters (cluster 1: groups 1, 2, 3, and 4; cluster 2: groups 5 and 6). According to the Spanish/non-Spanish classification, the first cluster of the UPGMA analysis (groups 1, 2, 3, and 4) included only 16 non-Spanish accessions out of 85. Group 1 included 15 non-Spanish and 13 Spanish accessions. The groups 2 and 3 did not show any non-Spanish, and the group 4 included only one, the ‘Jonadel’ accession. The second cluster (groups 5 and 6) was represented by 33 accessions, and 30 of them were characterized as non-Spanish accessions. The three Spanish accessions observed in this second cluster were ‘Biscarri-1’, ‘De Valdés’, and ‘Valsaina’ (Figure 4 and Figure 5).

The 19 phenotypic traits were divided into two groups of variables (Figure 4). Firstly, the TA, TPC, TFC, RAC, AsA, tartrate, oxalate, citrate, quinate, malate, and the sum of organic acids were clustered. On the other hand, succinate + shikimate, SSC, RI, sucrose, glucose, fructose, sorbitol, and the sum of individual sugars were grouped. Indeed, clusters would group variables that tended to behave similarly across the accessions. The first group of traits was dominated by the antioxidant parameters, the TA, and the individual organic acids (except the sum of shikimate and succinate), while the second corresponded to the individual sugars, the SSC, and the RI traits. Cluster 1 (groups 1, 2, 3, and 4) of accessions seemed to present a more acidic profile, while groups 5 and 6 showed a sweeter profile. Nevertheless, the more acidic cultivar assessed was non-Spanish (‘Astrakan Red’) and was in Cluster 1—group 1. The cluster 1 showed more antioxidant compound values than the groups 5 and 6. Indeed, the ‘Camuesa Fina de Aragón’ accession exhibited the highest TPC values with 98.1 mg GAE 100 g FW^−1^, far away from the first non-Spanish accession (‘Akane’) with 55.59 mg GAE 100 g FW^−1^. The ‘Les-1’ accession was the one with the highest relative antioxidant capacity (RAC) found among all the Pop4 data set (30.8 mg Trolox 100 g FW^−1^) while the ‘Delgared Infel’ (non-Spanish) was the one with the lowest RAC observed (5.9 mg Trolox 100 g FW^−1^). According to the harvest date, ranging from 170 (‘Mañaga-2’) to 316 (‘Bossost-4’), Julian days, the first cluster (groups 1, 2, 3, and 4) seemed to cluster accessions that ripened later than the accessions of the second cluster (groups 5 and 6). The peel color did not seem to influence clustering in the Pop4. Indeed, six accessions were characterized as ‘Red’ and they were in different groups: ‘Redaphough’ belonged to group number 1; ‘Averdal’, ‘Nueva Starking’, and ‘Starkrimson’ were in group 5, while ‘Delgared Infel’ and ‘Prima’ belonged to group 6.

### 3.4. Inter-Chromosomal Linkage Disequilibrium and Association Mapping

The inter-chromosomal linkage disequilibrium (LD) arrays of all 253 pairwise combinations of the 23 SSRs were assessed using TASSEL (Figure 6). The LD r^2^ values varied greatly from 0.00 to 0.77, and r^2^ = 1 indicates that two different markers provide exactly the same information. The highest LD value was recorded between the CH02c11 and CH02c06 markers. The significance cut-off threshold values from the distribution of LD were assigned at r^2^ = 0.2. Indeed, 15 pairwise combinations of the 23 SSRs assessed recorded values above the cut-off. Thus, CH01h10 and CH02c11 markers were considered to be in linkage disequilibrium, as well as markers such as CH02c06 with Hi02c07, CH05f06, CH02c09, and CH02c11; Ch02c09 with CH02c11; CH02c11 with CH05f06; MdSWEET2d with CH02c09, CH02c06, and CH02c11; and finally, MdSWEET7b with CH01h10, CH02c11, and CH02c06 and CH02c09 markers.

The results showed that 126 significant associations were reached between the 23 SSR markers and the 21 traits evaluated, using a modeling coefficient of membership (Q) value estimate from structure as a covariate (Table 5). In total, 20 SSRs (out of 23) contributing to the phenotypic variation were significantly associated with at least one trait studied. Moreover, only fructose was not associated with any of the 23 SSRs assessed. Furthermore, the SWEET gene markers MdSWEET2a, MdSWEET2d, MdSWEET7b, MdSWEET2b, MdSWEET7a, and MdSWEET12a showed associations (*p* ≤ 0.01) with 17 out of the 21 phenotypic traits assessed (harvest, peel color, RAC, TPC, TFC, SSC, TA, RI, sucrose, sorbitol, sum of sugars, oxalate, citrate, tartrate, malate, quinate, and sum of organic acids).

According to the agronomical parameters, five significant associations were found with the harvest date on LG5 (CH05f06 and MdSWEET2d), LG9 (CH01f03b), and LG14 (CH04c07 and MdSWEET12a), and two with the peel color and CH02c11 and MdSWEET7a markers on LG10 and 11. Regarding the antioxidant compounds, 23 associations were found between them, with 11 SSRs on LG2, 3, 4, 5, 6, 7, 10, 11, 12, 14, and 17. At a level of significance of *p* ≤ 0.0001, the relative antioxidant capacity (RAC) was associated with the MdSWEET7b marker, the TPC with the CH05f06 and MdSWEET7b markers, and finally, the TFC was linked with two markers (CH02c06 and CH05f06). For the basic fruit quality parameters, 25 associations were found between them, with 15 SSRs on LG1, 2, 3, 5, 6, 7, 9, 11, 12, 14, 15, and 17. At the level of significance (*p* ≤ 0.0001), the RI was associated with the CH04c07 marker and the TA with the CH01f02 marker. Concerning the sugar compounds, 15 associations have been observed between them, along with 10 SSR markers on LG1, 2, 4, 5, 11, 12, 13, 14, and 17. At the level of significance of *p* ≤ 0.0001, four significant associations were found between sucrose content and the CH-Vf1 marker, between sorbitol content and the CH01f02 and CH02c06 markers, and between the sum of the individual sugars and the MdSWEET12a marker. Lastly, according to the individual organic acids, 53 associations were found between them, with 17 SSRs in all LGs except the LG10 and LG13. At the level of significance *p* ≤ 0.0001, seven associations were found between the succinate and shikimate acids and the CH01f03b, CH02c06, and CH04e05 markers; the citric and quinic acids with the CH05f06 marker; and the oxalate acid with the CH02c06 and MdSWEET2a markers.

## 4. Discussion

### 4.1. Phenotypic Characterization

As expected, a large variability for the different traits assessed was found among accessions, and the concentrations of biochemical compounds observed in the present study were in line with previously reported values [18,27,53,54,55,56,57,58,59]. Nonetheless, the larger sample size of this study (118 accessions) resulted in a greater range of concentrations with significant differences in the values of the traits assessed.

Moreover, positive and significant correlations were found, as previously reported in other studies, between total phenolics and total flavonoids, or RAC [20,60,61]. Many of the correlations observed could be explained by photosynthetic activity [56,62]. Indeed, the photosynthetic products will act as substrates for many of the metabolic pathways. The sum of individual sugars and main products of the primary metabolism, photosynthesis and SSC, have shown significant and negative correlations with bioactive compounds (TPC and TFC) and positive correlations with all the organic acids, substrates, and products of the secondary metabolism in plants [63]. These correlations could thus be explained as a possible response to phenolic compound biosynthesis. Indeed, carbohydrates such as fructose increase the erythose-4-phosphate productions that, together with phosphoenolpyruvate (PEP), constitute a substrate for phenolic compounds through the shikimate pathway [64]. Furthermore, positive and significant correlations were also found between the antioxidants and the organic acids, as also reported by Mignard et al. [5] when they studied 155 apple accessions. The decarboxylation of dicarboxylates such as malate and other organic acids is linked to the degradation of organic acids, and thus, this decarboxylation permits PEP production. The PEP is closely associated with the activation of gluconeogenesis and thus results in glucose production in fruits [64]. These correlations showed the linkage between the primary and secondary metabolisms [65]. In fact, the secondary metabolism (antioxidant compounds) is connected to the primary metabolism (sugars and acids) as substrates are supplied from primary pathways and drove into the secondary biosynthetic routes. The positive and significant correlations between the different individual sugars could be explained through the polyol or sorbitol-aldose reductase pathway, a two-step mechanism converting glucose into fructose.

Moreover, high and significant correlations were found between the harvest date and the concentrations of SSC, sorbitol, sugars, and, as expected, with the RI. These results show that when fruits are harvested late in the season but each one is at its optimum commercial maturity, they are, in general, sweeter. In contrast, significant and negative correlations were found between the harvest date and individual/total organic acids, TA, the antioxidant compounds (TPC and TFC), and the relative antioxidant capacity (RAC) [65]. Furthermore, the non-Spanish accessions seemed to present lower organic acid concentrations and fewer antioxidant compounds (TPC, TFC, and RAC) compared with the Spanish well adapted accessions such as ‘Camuesa Fina de Aragón’, ‘Reguard–2’, ‘Transparente’, and ‘Verde Doncella’, as previously reported [5,20]. These results highlight the importance of germplasm characterization [66] with the aim of boosting autochthone accessions and phytogenetic resources for breeding programs.

### 4.2. Genetic Identity and Overall Diversity

The successful amplification and polymorphism obtained using 23 SSR markers covering the apple genome used to screen the 186 apple accessions (Pop1) confirmed previously reported results for cultivar identification and genetic mapping in apple trees [1,10,11,27,67,68,69,70]. 

Although some ancient apple Spanish accessions, relevant in the past in Spain [71], were assessed in this study, there was no proof that they were true-to-type accessions [1,11]. Indeed, the different accessions labeled ‘Camosa-1’/‘Camosa-2’, ‘Reguard-2’/‘Reguard-4’, ‘Mañaga-1’/‘Mañaga-2’ or ‘Verde Doncella-1’/‘Verde Doncella-3’ did not show the same SSR profiles. In contrast, ‘Verde Doncella-2’ and ‘Verde Doncella-3’ exhibited exactly the same allelic profiles. Moreover, the apple genetic resources analysis should be studied with attention because of the incidence of mutations, the genome structure variations, or the epigenetic alterations that could engender phenotypic modifications that are not distinguishable using only SSR markers [8,11,12]. Consequently, the phenotypic and molecular characterizations of apple accessions, both complementary approaches, could determine whether the accessions with the same SSR profiles would be the same accessions. 

Among the 23 different SSR markers, the level of genetic diversity and expected/observed heterozygosity were relatively high. As apple is a self-incompatible cross pollinating specie [11], the high diversity observed indicated that the apple genetic resources, and thus the accessions preserved in the germplasm bank of the EEAD–CSIC, were highly diverse. The high genetic diversity found was in agreement with the variability exhibited in other apple studies [1,8,11,23,68,70,72,73]. All SSR loci analyzed in this study displayed a high degree of polymorphism, with 4 to 35 alleles per locus and a 1.5 to 8.98 effective number of alleles per locus. Indeed, the mean value found in this study was a 15.65 number of alleles per locus for Pop1, which is slightly lower than the 16.69 observed by Urrestarazu et al. [1] studying 493 accessions and the 18.62 reported by Pereira-Lorenzo et al. [11] with 1453 accessions in Spain and sharing respectively 16 SSRs (13 markers out of 16 were used in the present study) and 13 SSR markers (12 out of 13 were included in the present work). Moreover, 18 alleles on average per locus were found in a Turkish germplasm collection of 206 accessions [67] and 19.5 in a collection of 2163 accessions in France [8]. Nevertheless, Marconi et al. [10], assessing 175 accessions in Italy, and Ferreira et al. [68], studying 87 accessions in Portugal, observed a mean value for the number of alleles per locus of 14.6 and 11.5 respectively. The observed heterozygosity averaged for the entire dataset (0.75) over the 23 SSR loci was exactly the same as reported by Meland et al. [74], studying 171 accessions and using seven SSR markers also used in the present study (CH02c06, GD12, CH01h10, CH02c11, CH02d08, CH02c09, and CH01h01 markers). Moreover, the H_0_ was higher than the 0.67 observed by Ferreira et al. [68] but slightly lower than reported values of 0.78 [10], 0.83 [8], 0.76 [67], and 0.81 [1]. The differences found in these studies could be due to the different plant material, the ploidy level, or even the number of accessions assessed in each study.

According to the triploid accessions, it is worth mentioning that the amplification of three alleles in a single locus is not evidence for characterizing an accession as triploid [23,68]. In fact, in diploid accessions, a third fragment could be amplified as the result of duplication or a somatic mutation [23]. However, if more than two alleles were found at several loci (several SSR markers), the accessions were thus characterized as triploids. Furthermore, in the present study, a confirmation of the ploidy level by flow cytometry for the whole dataset was assessed, as described by Reig et al. [18].

### 4.3. Population Structure

The structure results showed that there were two main subpopulations (K = 2) with some degree of admixture within both of them (Spanish and non-Spanish accessions). Similar studies in apples also reported two unstructured populations, indicating a strong subpopulation structure using respectively 1453 [11] and 493 accessions [1]. Indeed, Pereira-Lorenzo et al. [11] reported that the analysis of 1453 apple accessions, conserved in Spanish collections, permitted the discrimination of an Iberian genepool of apple accessions separated from an extensive set of non-Spanish reference modern cultivars.

The UPGMA cluster analysis grouped all the 118 apple accessions into a dendrogram. The cluster analysis was able to group all genotypes into two large clusters, with the first one (cluster 1) containing four subgroups (groups 1 to 4) and the second one (cluster 2) containing two subgroups (groups 5 and 6). The UPGMA analysis also showed that one of the two subpopulations encountered was quite exclusively formed by non-Spanish accessions (30 out of 33). Indeed, the cluster 1 included 16 non-Spanish cultivars (‘Astrakan Red’, ‘Baujade’, ‘Belleza de Roma’, ‘Blackjon’, ‘Bofla’, ‘Florina’, ‘Fuji’, ‘Granny Smith-1’, ‘Granny Smith-2’, ‘Idared’, ‘Jonadel’, ‘Jonathan-1’, ‘Jonathan-2’, ‘MacIntosh’, ‘Red Rome Beauty’, and ‘Redaphough’) and the cluster 2 included all the 30 remaining non-Spanish reference cultivars and only three Spanish accessions (‘Biscarri-1’, ‘De Valdés’, and ‘Valsaina’). Nevertheless, in the cluster 1, 15 non-Spanish accessions were included in the subgroup 1, and only one was in the subgroup 4 (‘Jonadel’). The groups 2 and 3 did not show any non-Spanish accession and could thus be referred to as the Iberian genepool described by Pereira-Lorenzo et al. [11].

The results for genetic subpopulation obtained suggested that the non-Spanish reference cultivars were slightly more similar among them than with the Spanish accessions. In fact, several studies have shown that the European germplasm core collections shared a lot of the non-Spanish reference plant material conserved [1,6,10,11,16,66,75,76]. Moreover, according to the autochthone accessions from the different germplasm collections around the world, domestication and breeding could have caused diversity loss. Nevertheless, regardless of the many decades of domestication of *Malus × domestica* and its clonal propagation, there is no proof that domesticated apples have shown a genetic bottleneck in contrast with *Malus × sieversii* [77,78].

It is interesting to note that the same Spanish/non-Spanish cluster segregation was also distinguishable in previous studies regarding the influence of climate parameters on basic fruit quality (SSC, TA, and RI), antioxidant traits (TPC, TFC, AsA, and RAC), individual sugars (glucose, fructose, sucrose, and sorbitol), and organic acids (citric, malic, oxalic, quinic, succinic, shikimic, and tartaric) contents of 155 accessions out of the 186 accessions included in Pop1 assessed in this study. Moreover, the range of the results was larger for the Spanish accessions than for the non-Spanish ones [5,20], highlighting more similar profiles according to the non-Spanish cultivars. Indeed, Spanish autochthone accessions reported a higher biodiversity and, in general, higher contents for some basic fruit quality traits, antioxidants, individual sugars, and organic acids than non-Spanish accessions over a period of five years of study. Higher contents of antioxidants and organic acids were observed, in general, in the groups 1, 2, 3, and 4 of the clustering analysis, while the sugar profile was more heterogeneous according to the accession’s origins. These results strengthen the awareness of the importance of autochthone phytogenetic resources and underline the high biodiversity found in germplasm core collections [66]. Moreover, climatic traits such as precipitation, solar radiation, and temperatures strongly influenced the antioxidant and metabolite profiles of the accessions studied and depended on their origin [5,18,20,55,56,65]. Nevertheless, Pereira-Lorenzo et al. [11] reported the same segregation with a total of 1453 accessions, including part of the accessions of the present work and different germplasm collections from Spain. Moreover, Mignard et al. [5,20] showed segregation between Spanish local accessions and modern non-Spanish cultivars according to the biochemical contents and the climatic factors influence (high solar radiation and low temperatures). This highlighted the fact that, further than the climatic parameters influence on the metabolite profiles of the apple fruits [79], the genetics have a stronger influence on the biochemical contents and explained why two subpopulations were found in previous fruit quality studies [5,20].

### 4.4. Association Mapping

To our knowledge, the present assay is the first study reported in apples concerning association mapping with agronomical and biochemical traits in a large germplasm collection using both Spanish and non-Spanish apple genetic resources. Twenty-one traits of interest, such as basic fruit quality, antioxidant parameters, individual sugars, and organic acids, were assessed by 23 SSR molecular markers. Regarding the bibliography, scarce studies of phenotypic associations in apples using SSRs can be found [80]. The most recent studies analyzed several parameters, such as flowering time, harvest date, flesh firmness, ripening index, or polyphenols, but based only on SNP markers [12,32,33,34,35,75,76]. However, Tsykun et al. [81] reported that the multi-allelic SSRs markers seemed to be best suited for detecting genetic structure than SNPs markers because the SSRs markers had a higher discrimination power than bi-allelic SNP markers [82].

Zhen et al. [27] reported five associations between the SWEET SSR markers and four individual sugars (sucrose, glucose, fructose, and sorbitol). Indeed, the MdSWEET2e marker was significantly associated with sucrose, fructose, and total sugars [27], but it was not in the present study. The MdSWEET9b marker showed significant association with the contents of sorbitol and was not linked significantly with fructose and total sugars, as reported by Zhen et al. [27]. Nevertheless, the MdSWEET9b marker also showed associations with RAC, oxalate, and quinate. In fact, Zhen et al. [27] did not find associations between the other SWEET genes SSR markers (MdSWEET7b, MdSWEET2d, MdSWEET2b, MdSWEET2a, MdSWEET12a, and MdSWEET17a) and the individual sugars. In the present study, more phenotypic traits were assessed, and thus, more significant associations were found.

Apart from the MdSWEET markers, the SSR markers assessed in the present work have not been used for association mapping with biochemical traits in previous studies. Nevertheless, QTLs with different phenotypic traits have been encountered, and these results may be very useful because many of the associated markers were located in common regions where major genes or QTLs have been previously identified on the apple genome [35,83,84,85,86,87,88,89].

According to the agronomical parameters, harvest date showed a significant association with the CH01f03b marker. Kenis et al. [86] reported a QTL with harvest date located in the LG9 as well as with the CH01f03b marker. Other studies [34], have shown associations between harvest date and chromosome 3, instead of LG9, using SNP markers, but they failed to find any candidate gene associated with this character. These differences in the results may be due to differences in plant material used and in the coverage density between SNPs and SSRs. The peel color trait was associated with the CH02c11 marker at LG10 and with the MdSWEET7a marker at LG11. Howard et al. [85] found many QTLs in LGs 2, 5, 8, and 9 regarding these agronomical traits. Moreover, Oh et al. [90] found QTLs for the apple skin color in LGs 9, 10, and 13 using the ‘Jonathan’ cultivar. A higher number of significant QTLs related to the basic fruit quality traits were found in previous works [84,86,87,89].

Significant associations between SSC and TA with the CH02c06 marker at LG2 were found. Guan et al. [84], Kunihisa et al. [87], and Zhang & Han [89] reported the same QTL for SSC in the LG2. Indeed, SSC was linked to the CH02c06 (LG2), the CH05f06 (LG5), the MdSWEET12a (LG14), and the CH02c09 marker (LG15), while TA was associated with SSR markers at LGs 1, 2, 3, 5, 9, 11, 12, and 14. Regarding the ripening index, the associations found by Urrestarazu et al. [12] in chromosomes 3, 10, and 16 were different from our outcomes. Fruit ripening trait is a complex character that is quantitatively inherited in most fruit tree species [91], and its control involves the regulation of many metabolic pathways such as starch, acidity, firmness, and changes in color, among others [92]. In any case, our results with SSRs would be a good approximation since Urrestarazu et al. [12] found two NAC transcription factors on LG3, close to the CH02c06 SSR marker on LG2, as candidate genes for the control of this trait. Indeed, the NAC genes constitute one of the largest families of plant-specific transcription factors and are present in a wide range of species [93]. In the same way, a strong micro-synteny was identified between *Malus* and *Prunus*, identifying a major locus on chromosome 4 controlling the maturity date of peaches. The NAC transcription factors controlling fruit ripening traits have also been described in tomato [94] and kiwifruit [95].

Antioxidant compounds were highly linked to several SSR markers assessed. Twenty six significant associations were found between the antioxidant compounds and SSR markers in LGs 2, 3, 4, 5, 6, 7, 10, 11, 12, 14, and 17. Chagné et al. [83] showed associations between antioxidant parameters and QTLs at LGs 3, 5, and 14, in agreement with those observed in the present study. In addition, McClure et al. [35] using GWAS identified a candidate gene (LAR1) within chromosome 16 for catechin production and several transcription factors of different classes (MYB, bHLH, bZIP, AP2). These results showed that the polyphenol content in apples may suggest that breeders may be able to improve the nutritional value of apples through marker assisted breeding (MAB) or gene editing. The high content of total phenols was correlated with high radiation and low temperatures, as shown by Mignard et al. [20]. These results are of vital importance, taking into account that extreme weather conditions will affect the fruit quality, with the Ebro Valley, in Spain, being one of the most vulnerable areas for apple production and also one of the most affected Mediterranean regions influenced by these changes.

According to the individual sugars, sucrose was highly linked to the CH-Vf1 marker (LG1), which is consistent with the findings of Larsen et al. [33] between sucrose and the chrl:30221387 SNP. Furthermore, these authors identified that VIN1 is linked to the chrl:30221387 SNP, making it a good candidate gene for this association. Other authors described different vacuolar invertase genes, including VIN1 [96], which play an important role in sugar metabolism in fruits. Further, Guan et al. [84] and Sun et al. [88] reported a QTL in the LG1 associated with sucrose and also other QTLs for fructose, glucose, and sorbitol but in different LGs.

Regarding the organic acids, a significant association with the CH04c07 marker (LG14) was found in the present work. Chagné et al. [83] reported a QTL in the LG14 linked to the quinic acid content. Moreover, for organic acids, significant associations between CH04e05 (LG7), CH01h10 (LG8), and CH02c09 markers (LG15) with oxalate, citrate, succinate, shikimate, and tartrate were found. Sun et al. [88] showed several QTLs in LGs 7, 8, and 15, which are the same LGs as the SSR markers linked to organic acids in the present study. Oh et al. [90] found a fruit acidity-related QTL in the LG13 of the ‘Jonathan’ cultivar, although no significant association was found in the present work.

The results obtained provided a strong base for further association mapping with agronomical and biochemical traits that could be applied to other species because of the synteny within the *Rosaceae* family [12,97,98]. Moreover, the 126 significant trait-marker associations found in the present study could provide potential information for effective marker assisted selection (MAS) in apple breeding programs. Despite being the first approximation made to date in apples between biochemical traits and SSR markers for the 21 parameters assessed, significant associations have been found and will be of great help in further work. New studies must be performed mapping thousands of SNPs (9K IRSC SNP arrays) to facilitate genome-wide scans and validate marker–locus–trait associations for application in breeding.

## 5. Conclusions

The present study provided new details about the EEAD-CSIC core collection population structure according to their ploidy level and their origin (Spanish/non-Spanish). The population structure analysis showed two subpopulations in agreement with the phenotypic segregation observed in previous studies. These results highlight the importance of understanding the genetic architecture of important fruit agronomical and biochemical traits because of the intrinsic correlation between the genetics and the metabolite profiles of the apple accessions. A total of 126 significant associations were observed between the 23 SSR markers assessed and the 21 phenotypic traits evaluated. These results would help in breeding programs with marker assisted selection (MAS) for fruit quality traits in apples, and in particular, the possible simultaneous selection for agronomical and biochemical parameters. The content of some nutritional compounds, such as total phenol content, also associated with some SSR markers, has turned out to be correlated with extreme climatic factors, making this a point to take into account in future fruit quality breeding programs. This study also identified many new marker-locus trait associations, such as with antioxidants or organic acid compounds, which may be useful for predictions and for a better understanding of the apple genome. Finding specific regions of the genome will provide further information regarding candidate genes involved in apple fruit quality.

### Future Perspectives

With the purpose of moving faster in the association mapping studies, future studies may be required in this area with a view to candidate gene identification and fine mapping using a large-scale of SNP markers. Recent advances in genomic tools such as genome wide association mapping (GWAS) and next-generation sequencing (NGS) have allowed the development of new approaches for mapping important and complex traits and facilitating SNP discovery. The identification of causal genes underlying specific traits is a major goal in plant breeding, subsequently offering opportunities to develop genomic selection tools. Moreover, due to the abundance of SNPs within a genome and the availability of high-throughput sequencing methods, SNPs are increasingly becoming one of the most commonly used markers for genotyping horticultural species.

## Figures and Tables

**Figure 1 plants-12-01249-f001:**
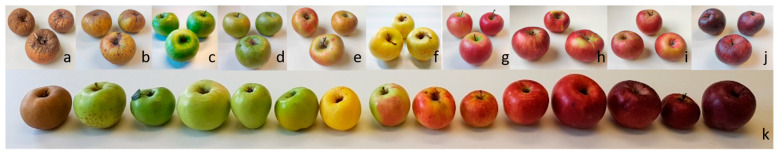
Phenotypic fruit diversity encompassed by the 186 accessions of the germplasm bank established at the EEAD-CSIC, Zaragoza, Spain. (**a**) ‘Reineta Gris’; (**b**) ‘Reineta Blanca del Canadá’; (**c**) ‘Granny Smith’; (**d**) ‘Baujade’; (**e**) ‘Morro de Liebre’; (**f**) ‘Golden Paradise’; (**g**) ‘Cripps Pink’; (**h**) ‘Reneta’; (**i**) ‘Solafuente’; (**j**) ‘Evasni’; (**k**) 15 different accessions from the EEAD-CSIC germplasm bank exhibiting differences in color and shape.

**Figure 2 plants-12-01249-f002:**
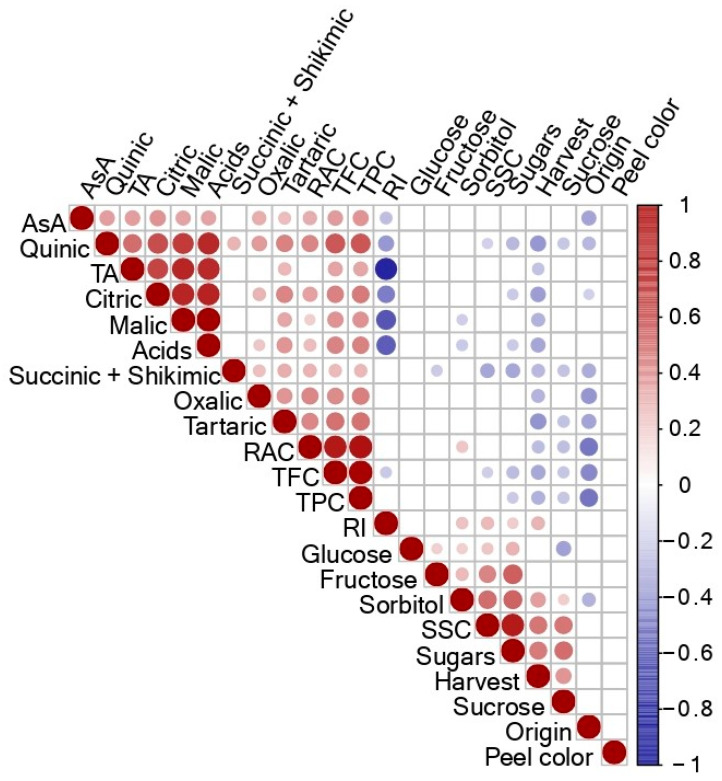
Pearson’s bivariate correlations among the different traits assessed over the 118 accessions phenotyped (Pop4). Abbreviations: SSC, soluble solids content; TA, titratable acidity; RI, ripening index; RAC, relative antioxidant content; TPC, total phenolics content; TFC, total flavonoids content; AsA, Ascorbic acid; Sugars, sum of individual sugars; Acids, sum of individual organic acids. The size of the circle for each correlation and the colour depicts the significance and the magnitude of the correlation coefficient, respectively.

**Figure 3 plants-12-01249-f003:**
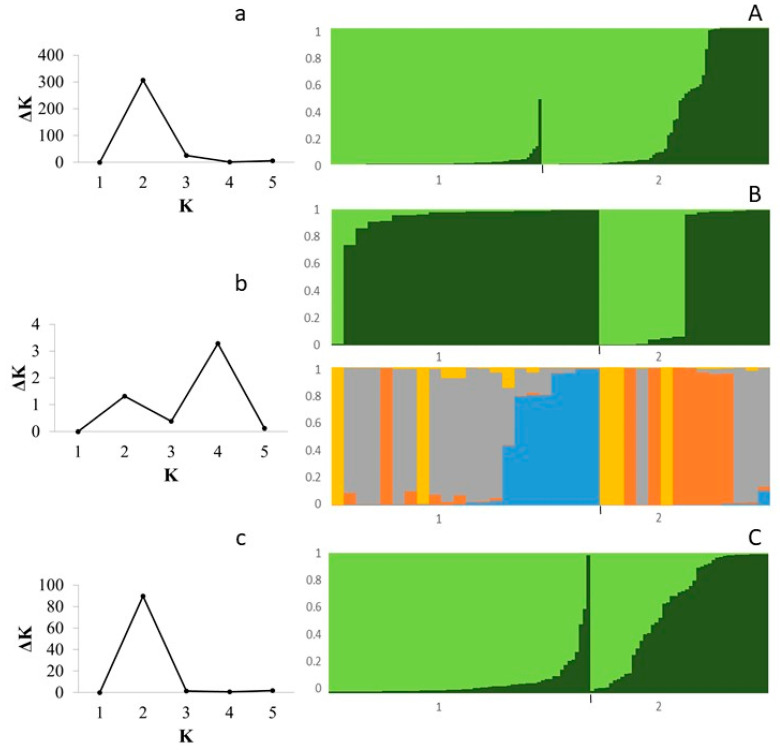
Estimation of the *Malus × domestica* Borkh collection using LnP (D) derived ΔK for K from 1 to 5 based on: (**a**) Pop2 (150 diploids apple accessions); (**b**) Pop3 (36 triploids); and (**c**) Pop4 (118 diploids) and STRUCTURE bar plots based on: (**A**) Pop2 at K = 2; (**B**) Pop3 at K = 2 and K = 4; and (**C**) Pop4 at K = 2, sorting by subpopulation (Spanish/Non-Spanish) and the coefficient of membership (Q).

**Figure 4 plants-12-01249-f004:**
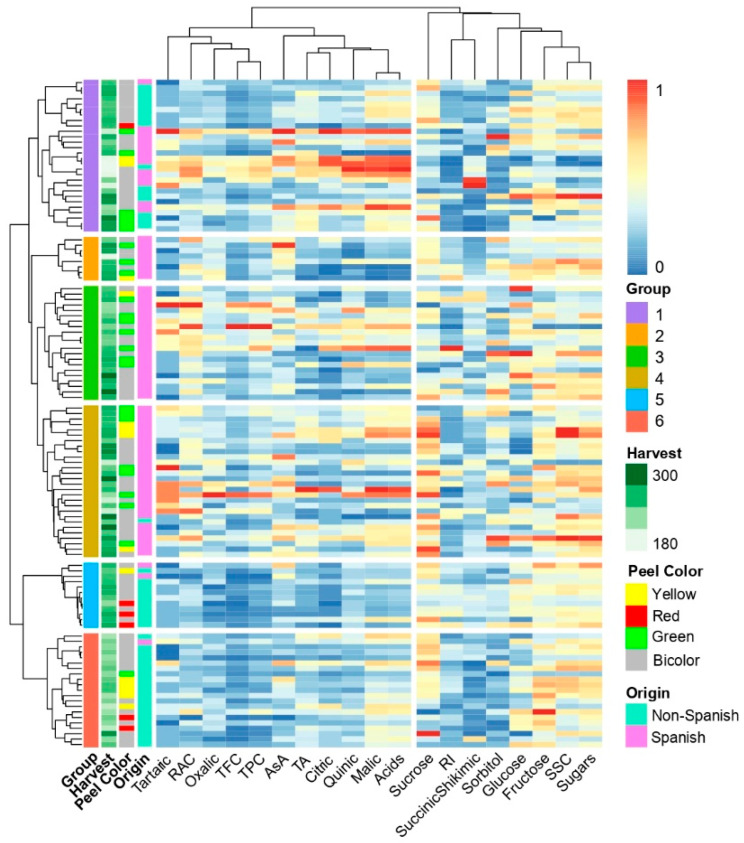
Pheatmap of 118 apple accessions (Pop4) based on pairwise genetic distances with 23 SSR markers and fruit biochemical characteristics as basic fruit quality (soluble solids content, SSC; titratable acidity, TA; ripening index, RI), antioxidants (total phenolics content, TPC; total flavonoids content, TFC; vitamin C-AsA), individual sugars (glucose, fructose, sucrose and sorbitol), and organic acids (malate, quinate, citrate, tartrate, oxalate, succinate, and shikimate). Annotations as group (*n* = 6 groups), harvest date, peel color and origin (Spanish/Non-Spanish) for each accession are shown.

**Figure 5 plants-12-01249-f005:**
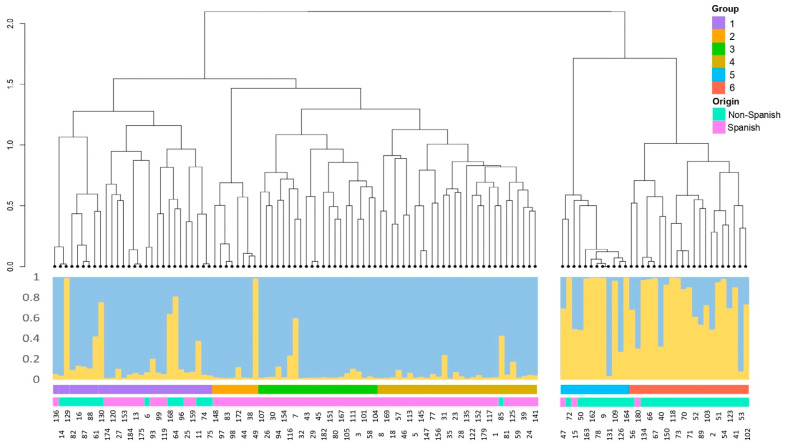
Dendrogram of Pop4, corresponding to 118 diploids apple accessions, based on pairwise genetic distances according to 23 SSR markers and STRUCTURE bar plot at K = 2, optimum number of K subpopulations for Pop4. Annotations as group (*n* = 6 groups) and origin (Spanish/Non-Spanish) for each accession are shown.

**Figure 6 plants-12-01249-f006:**
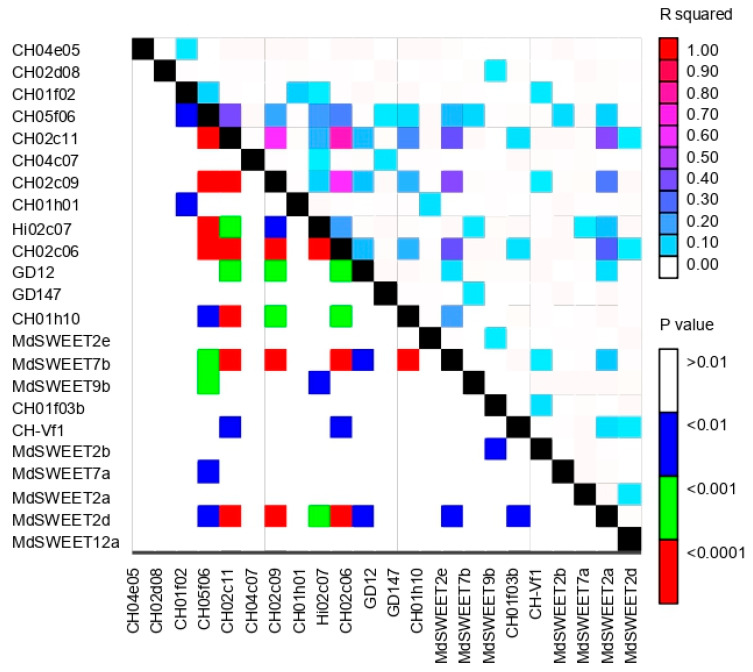
Inter-chromosomal linkage disequilibrium (LD) plot based on 23 SSR markers screened among Pop4 (118 apple accessions). The r^2^ values are shown in the upper right of the plot and the *p*-values are represented in lower left of the plot.

**Table 1 plants-12-01249-t001:** Basic information of the 186 apple accessions (Pop1) assessed in this study.

N°	Accession	EEAD Code	Ploidy	Origin
1	Aciprés *	3339	2n	Spanish
2	Akane *	2902	2n	non-Spanish
3	Almenar-2 *	3555	2n	Spanish
4	Ascara-1	3423	3n	Spanish
5	Ascara-2 *	3424	2n	Spanish
6	Astrakan Red *	3378	2n	non-Spanish
7	Audiena de Oroz *	3375	2n	Spanish
8	Augüenta *	3335	2n	Spanish
9	Averdal-1 *	882021	2n	non-Spanish
10	Averdal-2	892340	2n	non-Spanish
11	Baujade *	923284	2n	non-Spanish
12	Belgolden	3193	2n	non-Spanish
13	Bellaguarda Lardero *	3547	2n	Spanish
14	Belleza de Roma *	638	2n	non-Spanish
15	Biscarri-1 *	3726	2n	Spanish
16	Blackjon *	2690	2n	non-Spanish
17	Blacktayman	2490	3n	non-Spanish
18	Bofla *	3418	2n	Spanish
19	Boluaga	3340	3n	Spanish
20	Boskoop Rouge	2898	3n	non-Spanish
21	Bossost-1	3626	3n	Spanish
22	Bossost-2	3627	3n	Spanish
23	Bossost-4 *	3629	2n	Spanish
24	Bost Kantoia *	3341	2n	Spanish
25	Cabdellà-2 *	3613	2n	Spanish
26	Cabello de Angel *	3255	2n	Spanish
27	Calvilla de San Salvador *	3342	2n	Spanish
28	Camosa-1 *	3553	2n	Spanish
29	Camosa-2 *	3620	2n	Spanish
30	Camuesa de Daroca *	3371	2n	Spanish
31	Camuesa de Llobregat *	1342	2n	Spanish
32	Camuesa Fina de Aragón *	3372	2n	Spanish
33	Carapanón	3634	3n	Spanish
34	Carrió	3636	3n	Spanish
35	Cella *	2512	2n	Spanish
36	Cepiland	881967	2n	non-Spanish
37	Charden	303	3n	non-Spanish
38	Ciri Blanc *	3402	2n	Spanish
39	Cirio *	3615	2n	Spanish
40	Cox’s Orange Pippin *	2889	2n	non-Spanish
41	Cripps Pink *	933540	2n	non-Spanish
42	Crispin	3080	3n	non-Spanish
43	Cuallarga *	3467	2n	Spanish
44	Cul de Cirio *	3551	2n	Spanish
45	De Agosto *	3619	2n	Spanish
46	De Pera *	3416	2n	Spanish
47	De Valdés *	3632	2n	Spanish
48	Delbar Estivale	3262	2n	non-Spanish
49	Delciri *	3413	2n	Spanish
50	Delcon *	2896	2n	non-Spanish
51	Delgared Infel *	902708	2n	non-Spanish
52	Deljeni *	851305	2n	non-Spanish
53	Delkistar *	923273	2n	non-Spanish
54	Delorgue Festival *	913044	2n	non-Spanish
55	Democrat	2892	3n	non-Spanish
56	Elista *	912883	2n	non-Spanish
57	Esperiega *	3420	2n	Spanish
58	Esperiega de Olba *	3725	2n	Spanish
59	Eugenia *	3468	2n	Spanish
60	Evasni (Scarlet Spur)	933554	2n	non-Spanish
61	Florina *	3633	2n	non-Spanish
62	Fortuna Delicious	2702	2n	non-Spanish
63	Freyberg	2611	2n	non-Spanish
64	Fuji *	3488	2n	non-Spanish
65	Fukutami	2895	2n	non-Spanish
66	Gala *	3197	2n	non-Spanish
67	Galaxy *	892451	2n	non-Spanish
68	Gloster 69	3140	2n	non-Spanish
69	Golden Auvil Spur	2402	2n	non-Spanish
70	Golden Delicious *	675	2n	non-Spanish
71	Golden Delicious Infel *	2491	2n	non-Spanish
72	Golden Paradise *	3739	2n	non-Spanish
73	Golden Smoothee *	3286	2n	non-Spanish
74	Granny Smith-1 *	3196	2n	non-Spanish
75	Granny Smith-2 *	2614	2n	non-Spanish
76	Gravenstein	3109	3n	non-Spanish
77	Guillemes *	3411	2n	Spanish
78	Hared *	892232	2n	non-Spanish
79	Harrold Red Delicious	2899	2n	non-Spanish
80	Helada *	3368	2n	Spanish
81	Hierro *	3374	2n	Spanish
82	Idared *	2484	2n	non-Spanish
83	Irgo-2 *	3622	2n	Spanish
84	Jerseymac	3141	2n	non-Spanish
85	Jonadel *	2650	2n	non-Spanish
86	Jonagored	882001	3n	non-Spanish
87	Jonathan-1 *	2495	2n	non-Spanish
88	Jonathan-2 *	3096	2n	non-Spanish
89	Jubilee *	851304	2n	non-Spanish
90	Kidd’s Orange Red	2888	2n	non-Spanish
91	Kinrei	2900	2n	non-Spanish
92	Lancer	881968	2n	non-Spanish
93	Landetxo *	3343	2n	Spanish
94	Les-1 *	3624	2n	Spanish
95	Les-2	3625	3n	Spanish
96	MacIntosh *	3192	2n	non-Spanish
97	Mañaga-1 *	469	2n	Spanish
98	Mañaga-2 *	3554	2n	Spanish
99	Marinera *	3412	2n	Spanish
100	Marquiñez	3419	3n	Spanish
101	Médulas-1 *	3548	2n	Spanish
102	Melrose *	2484	2n	non-Spanish
103	Merrigold *	851307	2n	non-Spanish
104	Montcada-1 *	3631	2n	Spanish
105	Morro de Liebre *	3256	2n	Spanish
106	Mutsu	2487	3n	non-Spanish
107	Nesple *	3410	2n	Spanish
108	Normanda	3252	3n	Spanish
109	Nueva Starking *	1899	2n	non-Spanish
110	Ortell-1	413	3n	Spanish
111	Ortell-2 *	3546	2n	Spanish
112	Ozark Gold	3175	2n	non-Spanish
113	Pera 2 *	3417	2n	Spanish
114	Pera de Sangüesa	3379	3n	Spanish
115	Pero Pardo	3369	3n	Spanish
116	Peromingan *	1158	2n	Spanish
117	Peruco de Caparroso *	3373	2n	Spanish
118	Plaona *	923283	2n	non-Spanish
119	Poma de San Juan *	3556	2n	Spanish
120	Prau Riu-3 *	3491	2n	Spanish
121	Prau Riu-4	3492	3n	Spanish
122	Prau Riu-5 *	3493	2n	Spanish
123	Prima *	851306	2n	non-Spanish
124	Prime Gold	3198	2n	non-Spanish
125	Rebellón *	3370	2n	Spanish
126	Red Delicious *	3085	2n	non-Spanish
127	Red Elstar	882002	2n	non-Spanish
128	Red King Delicious	2688	2n	non-Spanish
129	Red Rome Beauty *	2897	2n	non-Spanish
130	Redaphough *	933411	2n	non-Spanish
131	RedChief *	851308	2n	non-Spanish
132	Redspur Delicious	3082	2n	non-Spanish
133	Regal Prince-1	882022	2n	non-Spanish
134	Regal Prince-2 *	892341	2n	non-Spanish
135	Reguard-2 *	3617	2n	Spanish
136	Reguard-4 *	3618	2n	Spanish
137	Reina de Reinetas	2488	3n	non-Spanish
138	Reineta Blanca del Canadá-1	308	3n	non-Spanish
139	Reineta Blanca del Canadá-2	3111	3n	non-Spanish
140	Reineta Blanca del Canadá-3	3194	3n	non-Spanish
141	Reineta Encarnada *	3635	2n	Spanish
142	Reineta Gris	2883	3n	non-Spanish
143	Reineta Inesita Asua	2543	3n	Spanish
144	Reineta Regil	3466	3n	Spanish
145	Reneta *	3408	2n	Spanish
146	Richared Delicious	2481	2n	non-Spanish
147	Roja Valle de Benejama *	1038	2n	Spanish
148	Roser de la Reula *	3552	2n	Spanish
149	Royal Red Delicious	2363	2n	non-Spanish
150	Rubinete *	861526	2n	non-Spanish
151	Ruixou-1 *	3614	2n	Spanish
152	San Felipe *	3376	2n	Spanish
153	San Miguel *	2579	2n	Spanish
154	Sandía *	3336	2n	Spanish
155	Sant Jaume	3470	3n	Spanish
156	Sant Joan *	3409	2n	Spanish
157	Santa Margarida	3401	3n	Spanish
158	Shelred	2893	2n	non-Spanish
159	Signatillis *	3403	2n	Spanish
160	Solafuente	3559	3n	Spanish
161	Spartan	2483	2n	non-Spanish
162	Starking-1 *	2964	2n	non-Spanish
163	Starking-2 *	632	2n	non-Spanish
164	Starkrimson-1 *	3195	2n	non-Spanish
165	Starkrimson-2	1904	2n	non-Spanish
166	Stayman Waynesap	3110	3n	non-Spanish
167	Taüll-1 *	3623	2n	Spanish
168	Telamon *	3398	2n	non-Spanish
169	Tempera *	3334	2n	Spanish
170	Terrera	3469	3n	Spanish
171	Topred Delicious	2651	2n	non-Spanish
172	Totxa *	3471	2n	Spanish
173	Trajan	3396	2n	non-Spanish
174	Transparente *	3377	2n	Spanish
175	Transparente Blanca *	3344	2n	Spanish
176	Turley Winnesap	2884	3n	non-Spanish
177	Tuscan	3397	2n	non-Spanish
178	Urarte	3415	3n	Spanish
179	Urtebete *	3345	2n	Spanish
180	Valsaina *	3558	2n	Spanish
181	Vance Delicious	2647	2n	non-Spanish
182	Verde Doncella-1 *	2125	2n	Spanish
183	Verde Doncella-2	310	2n	Spanish
184	Verde Doncella-3 *	3549	2n	Spanish
185	Vinçada Tardía	3621	3n	Spanish
186	Wellspur Delicious	3081	2n	non-Spanish

* Accessions assessed for the association study.

**Table 2 plants-12-01249-t002:** Characteristics of the 26 SSR markers used in this study with indication of the corresponding multiplex and dye.

Locus	LG	Multiplex	Dye	Size Range (bp)	Forward Primer Sequence (5′→3′)	Reverse Primer Sequence (5′→3′)	Primer Concentration	Reference
CH-Vf1	1	MP_5_	VIC	130–171	ATCACCACCAGCAGCAAAG	CATACAAATCAAAGCACAACCC	[0.1 µM]	[1]
Hi02c07	1	MP_3_	VIC	98–146	AGAGCTACGGGGATCCAAAT	GTTTAAGCATCCCGATTGAAAGG	[0.1 µM]	[1]
CH02c06	2	MP_3_	PET	201–261	TGACGAAATCCACTACTAATGCA	GATTGCGCGCTTTTTAACAT	[0.4 µM]	[1]
GD12	3	MP_3_	NED	139–189	TTGAGGTGTTTCTCCCATTGGA	CTAACGAAGCCGCCATTTCTTT	[0.1 µM]	[1]
MdSWEET2a	3	MP_6_	VIC	331–357	ATACCGAGGAACTGTAGGACCAAGC	CTCCACACTAAACAACCAGAAAGCA	[0.1 µM]	[27]
MdSWEET9b	4	MP_4_	6-FAM	336–360	GCGCCAATGTAAGACCCTTTACTTT	CTGACCTTGTCCTTCTTGGATGCGTA	[0.1 µM]	[27]
CH05f06	5	MP_2_	NED	161–189	TTAGATCCGGTCACTCTCCACT	TGGAGGAAGACGAAGAAGAAAG	[0.1 µM]	[1]
MdSWEET2d	5	MP_6_	PET	265–289	CATTCAATTTATTCGACCGGACGAC	TGGGTTCATCCCTCACTTTCACTCA	[0.1 µM]	[27]
MdSWEET7b	6	MP_4_	VIC	230–267	GGGTTTTGAGAATCTTGAGGGTAGG	TTTGATGGGTTGGACTGTAACTTGC	[0.1 µM]	[27]
CH03d07	6	MP_3_	VIC	n.a.	CAAATCAATGCAAAACTGTCA	GGCTTCTGGCCATGATTTTA	[0.1 µM]	[1]
CH04e05	7	MP_1_	PET	163–228	AGGCTAACAGAAATGTGGTTTG	ATGGCTCCTATTGCCATCAT	[0.1 µM]	[1]
CH01h10	8	MP_4_	PET	81–120	TGCAAAGATAGGTAGATATATGCCA	AGGAGGGATTGTTTGTGCAC	[0.1 µM]	[1]
CH01f03b	9	MP_5_	NED	127–177	GAGAAGCAAATGCAAAACCC	CTCCCCGGCTCCTATTCTAC	[0.1 µM]	[1]
CH01h02	9	MP_1_	NED	n.a.	AGAGCTTCGAGCTTCGTTTG	ATCTTTTGGTGCTCCCACAC	[0.1 µM]	[1]
CH02c11	10	MP_2_	PET	208–238	TGAAGGCAATCACTCTGTGC	TTCCGAGAATCCTCTTCGAC	[0.15 µM]	[1]
MdSWEET2e	10	MP_4_	NED	205–243	GTGAGCCCACAACTAATCCCAT	CTTGTGCGTAGGAATCCCGATA	[0.1 µM]	[27]
CH02d08	11	MP_1_	VIC	196–256	TCCAAAATGGCGTACCTCTC	GCAGACACTCACTCACTATCTCTC	[0.1 µM]	[1]
MdSWEET2b	11	MP_5_	6-FAM	249–263	TGAGGCAGAAACAATCATAAGGGTC	GAGCACGGAATTTGAAGCTGTAAAA	[0.1 µM]	[27]
MdSWEET7a	11	MP_5_	PET	340–376	TTCTATCTCCCCTTCCCAAACTTCC	GCTAAACAGTGCCACTGCATAAGGT	[0.1 µM]	[27]
CH01f02	12	MP_1_	6-FAM	155–212	ACCACATTAGAGCAGTTGAGG	CTGGTTTGTTTTCCTCCAGC	[0.1 µM]	[1]
GD147	13	MP_3_	PET	124–158	TCCCGCCATTTCTCTGC	GTTTAAACCGCTGCTGCTGAAC	[0.1 µM]	[1]
CH04c07	14	MP_2_	VIC	93–139	GGCCTTCCATGTCTCAGAAG	CCTCATGCCCTCCACTAACA	[0.1 µM]	[1]
MdSWEET12a	14	MP_6_	NED	223–253	ATGACAGGGCAACTTCAGGGT	CGTAATAGTCCTTTGCCCTCC	[0.1 µM]	[27]
CH02c09	15	MP_2_	VIC	203–254	TTATGTACCAACTTTGCTAACCTC	AGAAGCAGCAGAGGAGGATG	[0.1 µM]	[1]
CH04f10	16	MP_3_	6-FAM	n.a.	GTAATGGAAATACAGTTTCACAA	TTAAATGCTTGGTGTGTTTTGC	[0.1 µM]	[1]
CH01h01	17	MP_2_	6-FAM	92–130	GAAAGACTTGCAGTGGGAGC	GGAGTGGGTTTGAGAAGGTT	[0.1 µM]	[1]

Abbreviations: n.a. = no or weak amplification.

**Table 3 plants-12-01249-t003:** Basic statistics of phenotypical traits over the Pop4 (118 diploid accessions) during the 2014–2018 period: units, minimum, maximum, mean values, standard deviation, and standard error of the mean.

Trait	Units	Minimum	Maximum	Mean	SD	SE	ANOVA
SSC	°Brix	10.14	17.03	13.40	1.36	0.47	***
TA	g malic acid L^−1^	1.77	17.29	6.61	2.92	1.35	***
RI	-	0.76	8.55	2.62	1.34	1.45	***
TPC	mg GAE 100 g FW^−1^	15.24	98.07	39.54	15.76	0.08	***
TFC	mg CE 100 g FW^−1^	6.00	88.95	22.84	14.67	0.12	***
AsA	mg AsA 100 g FW^−1^	1.37	5.30	2.83	0.82	0.27	***
RAC	mg Trolox 100 g FW^−1^	5.93	30.82	15.44	5.08	0.12	***
Sucrose	g.kg^−1^	10.29	42.14	25.52	7.01	0.64	***
Glucose	g.kg^−1^	6.23	24.29	13.17	4.12	0.38	***
Fructose	g.kg^−1^	31.39	61.41	45.55	5.17	0.48	***
Sorbitol	g.kg^−1^	1.20	11.96	4.55	2.43	0.22	***
Sugars	g.kg^−1^	63.69	115.27	88.76	9.87	0.91	***
Oxalic	g.kg^−1^	0.0136	0.0176	0.0147	0.0006	0.0001	***
Citric	g.kg^−1^	0.0178	0.1482	0.0556	0.0279	0.0026	***
Tartaric	g.kg^−1^	0.0260	0.0910	0.0480	0.0146	0.0013	***
Malic	g.kg^−1^	2.6831	9.8144	5.5182	1.6930	0.1559	***
Quinic	g.kg^−1^	0.2350	0.8005	0.4231	0.1151	0.0106	***
Succinic + Shikimic	g.kg^−1^	0.1948	1.6766	0.5570	0.2520	0.0232	***
Acids	g.kg^−1^	3.4242	11.3889	6.6085	1.8218	0.1677	***

ANOVA: significant differences at *p* ≤ 0.001 (***) among the different apple accessions. Abbreviations: SSC, soluble solids content; TA, titratable acidity; RI, ripening index; TPC, total phenolics content; TFC, total flavonoids content; AsA, Ascorbic acid; RAC, relative antioxidant content; Sugars, sum of individual sugars; Acids, sum of organic acids.

**Table 4 plants-12-01249-t004:** Mean estimated values for different genetic parameters of the 186 apple accessions (Pop1) based on 23 SSRs loci.

SSR	N_A_	N_E_	N_B_	Ho	He	F_is_
CH-Vf1	13	3.50	10	0.75	0.73	−0.03
Hi02c07	9	3.40	4	0.68	0.70	0.03
CH02c06	21	8.54	14	0.93	0.90	−0.03
GD12	17	2.85	13	0.71	0.69	−0.03
MdSWEET2a	12	4.28	8	0.83	0.81	−0.02
MdSWEET9b	9	2.53	5	0.66	0.63	−0.05
CH05f06	14	5.79	7	0.86	0.92	0.07
MdSWEET2d	13	4.88	8	0.44	0.81	0.46
MdSWEET7b	15	5.17	9	0.96	0.87	−0.10
CH04e05	30	3.47	26	0.67	0.73	0.08
CH01h10	18	3.23	14	0.78	0.72	−0.08
CH01f03b	15	5.24	11	0.92	0.82	−0.12
CH02c11	15	8.72	5	0.89	0.90	0.01
MdSWEET2e	12	1.50	10	0.19	0.41	0.54
CH02d08	25	8.98	18	0.85	0.90	0.06
MdSWEET2b	4	1.96	2	0.51	0.50	−0.02
MdSWEET7a	12	3.61	8	0.72	0.74	0.03
CH01f02	35	8.60	29	0.81	0.90	0.10
GD147	15	4.29	9	0.83	0.78	−0.06
CH04c07	19	7.06	11	0.92	0.91	−0.01
MdSWEET12a	8	2.42	4	0.60	0.58	−0.03
CH02c09	13	6.96	5	0.93	0.87	−0.07
CH01h01	16	5.80	10	0.75	0.83	0.10
Mean-186 accessions (Pop1)	15.65	4.90	10.43	0.75	0.77	0.03
Mean diploids-150 accessions (Pop2)	14.70	5.23	9.22	0.79	0.77	−0.03
Mean triploids-36 accessions (Pop3)	10.00	4.66	5.26	0.71	0.73	0.03

N_A_: observed number of alleles per locus; N_E_: effective number of alleles per locus; N_B_: number of rare alleles; Ho: observed heterozygosity; He: expected heterozygosity, F_is_: Wright’s fixation index; I: Shannon’s information index; Pop: population.

**Table 5 plants-12-01249-t005:** Significance (*p-value*) of association between 23 SSRs polymorphic loci and biochemical traits in 118 apple accessions (Pop4).

		Agronomical		Antioxidants				Basic Fruit Quality			Individual Sugars					Individual Organic Acids						
LG	Marker Name	Harvest	Peel Color	RAC	TFC	TPC	AsA	SSC	TA	RI	Suc	Glu	Fru ^1^	Sor	Sug	Oxa	Cit	Tar	Mal	Qui	Succ-Shi	Acids
1	CH-Vf1										***							*			**	
1	Hi02c07								**	**							*		*			
2	CH02c06				***	**		*	*	*	**	*		***	*	***	*	*		*	***	
3	GD12 ^1^																					
3	MdSWEET2a			**	*	*			*							***	*	**	*	*		*
4	MdSWEET9b			*										*		**				*		
5	CH05f06	*		**	***	***		**	**	*					*		***	*	*	***	**	**
5	MdSWEET2d	*				*			**								**		**	*		**
6	MdSWEET7b			***	**	***				*								*				
7	CH04e05						**			**							*				***	
8	CH01h10															**						
9	CH01f03b	*							**	*							**		**	*	***	**
10	CH02c11		**	**		*	*															
10	MdSWEET2e ^1^																					
11	CH02d08 ^1^																					
11	MdSWEET2b								*							*	*	*	*	*		*
11	MdSWEET7a		*			*			**	*					*		**					
12	CH01f02			*		**			***	*				***			*					*
13	GD147													**								
14	CH04c07	*		*	*	**	*		**	***		*				*	*		*	*	*	*
14	MdSWEET12a	***						**			*			**	***							
15	CH02c09							*		*								**				
17	CH01h01			*		*				*	*							**				

^1^ No association found for this SSR or trait. Statistical significance at *: *p* ≤ 0.01; **: *p* ≤ 0.001; ***: *p* ≤ 0.0001; Abbreviations: RAC, relative antioxidant content; TFC, total flavonoids content; TPC, total phenolics content; AsA, Ascorbic acid; SSC, soluble solids content; TA, titratable acidity; RI, ripening index; Suc, sucrose; Glu, glucose; Fru, fructose; Sor, sorbitol; Sug, total sugars; Oxa, oxalate; Cit, citrate; Tar, tartrate; Mal, malate; Qui, Quinate; Succ-Shi, succinate and shikimate; Acids, total organic acids.

## Data Availability

Not applicable.

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
