# Peer review of "Population Structure and Association Mapping for Agronomical and Biochemical Traits of a Large Spanish Apple Germplasm"

_plants, 2023, doi:10.3390/plants12061249_

Round 1

Reviewer 1 Report

P1: banana is tropical fruit crop so this sentence needs to be revised.

P2: Domesticated apple is considered to be derived from autopolyploidization of a Gillenia-like taxon followed by diploidization (Velasco et al., 2010). The sentence related to inter-specific hybrid needs to be revised.

P3: Malus x domestica, remove italic style for the crossing mark. There is other parts needs to be modified.

P8: Ripening index.  Are there any papers used to assess ripening according to this criteria? As total sugars and acidity varies among cultivars, I wonder if this index can be used for ripening index. Simply indicating sugar-acid ratio may be good.

P18, P20: UPGMA analysis. Some cultivars showed incongruity for the result of clustering and their origin (Spanish/non-spanish). Do you explain the incongruity based on their parentage, since some cultivars are likely derived from crossing between the two population.

P21: In my opinion, LD refers to genetic linkage of the loci located on the same chromosome. However, authors showed LD values of SSR markers which are located on different chromosome (CH02c11 on LG10, CH02c06 on LG2). This issue needs to be addressed.

Table2: Previous QTL analysis indicates that significant QTLs for acidity and polyphenol content are located on LG16, but there is no SSR marker from LG16 in this study. To do association mapping, markers should cover all chromosomes. Please clarify this point.

Table5: This result only indicates statistical significance for association in a given marker and a trait. For breeding purpose, we want to know degree of association in terms of phenotypic variation. This reviewer asks to include additional data , which shows relationship of marker genotype and phenotypic variation of a trait.

Author Response

Comments from Reviewer 1:

P1: banana is tropical fruit crop so this sentence needs to be revised.

We appreciate this reviewer’s suggestion and agree with his comment. Changes were consequently done and the final sentence in the revised manuscript is: ‘Modern cultivated apple (Malus x domestica Borkh.) is the most significant and ancient fruit crop of the Rosaceae in the world as well as in Spain (Urrestarazu et al., 2012)’.

P2: Domesticated apple is considered to be derived from autopolyploidization of a Gillenia-like taxon followed by diploidization (Velasco et al., 2010). The sentence related to inter-specific hybrid needs to be revised.

We appreciate and agree with the reviewer’s comment. Changes were consequently done and a reference has been added: Gross et al., 2012. The sentence in the revised manuscript is: ‘All commercial apple cultivars descend from an inter-specific hybrid complex belonging to the Rosaceae family, having a basic chromosome number of 17 after an auto-polyploidization of a Gillenia-like taxon followed by diploidization (Gross et al., 2012; Marconi et al., 2018; Velasco et al., 2010)’.

P3: Malus x domestica, remove italic style for the crossing mark. There is other parts needs to be modified.

We agree with the suggested modification, the comment has been taken into account and the text has been thus revised entirely.

P8: Ripening index. Are there any papers used to assess ripening according to this criteria? As total sugars and acidity varies among cultivars, I wonder if this index can be used for ripening index. Simply indicating sugar-acid ratio may be good.

We agree with the reviewer comment and we would like to share that there are many papers using the SSC/TA ratio as the corresponding ripening index of the fruit. Nevertheless, as the reviewer has indicated, as total sugars and acidity varies among cultivars, we did not use this index as an indicator of general ripening for defining harvest time. The ripening index is thus, as commented by the reviewer, used as the sugar-acid ratio although this ratio is also known as ripening index in many articles.

P18, P20: UPGMA analysis. Some cultivars showed incongruity for the result of clustering and their origin (Spanish/non-Spanish). Do you explain the incongruity based on their parentage, since some cultivars are likely derived from crossing between the two populations?

We would thank the reviewer for this comment. In fact, apples genetic reproduction excludes self-fertilization and, this self-incompatibility is a way of insuring that each apple seed will be a hybrid between the maternal (seed) parent and the paternal (pollen) parent. The Spanish accessions ‘De Valdes’, ‘Biscarri’, and ‘Valsaina’ were in the second cluster of the UPGMA cluster analysis, which is quite exclusively formed by non-Spanish cultivars. Nevertheless, the parentage of these three Spanish accessions is unknown and thus, if one of the parents of these accessions was a non-Spanish cultivar it is also unknown. If one of the parents was non-Spanish, the incongruity of these accessions could be explained by their parentage but, the incongruity could also be explained by many other explications as self-mutation for instance. However, the UPGMA cluster analysis assessed in the present work showed a clear division between the Spanish and the non-Spanish accessions evaluated. Indeed, the Cluster 2 grouped 30 non-Spanish cultivars out of 33; and the Cluster 1, formed by four different groups, exhibited 16 non-Spanish cultivars out of 85 total accessions. For instance, in the Cluster 1 the ‘Jonadel’ accession is included and its parentage is already known (‘Jonathan’ x ‘Red Delicious’) but both parents are non-Spanish and thus, the parentage did not explain this incongruity. Nevertheless, for the other 15 non-Spanish accessions of the Cluster 1, the parentage could or not explain the incongruity.

P21: In my opinion, LD refers to genetic linkage of the loci located on the same chromosome. However, authors showed LD values of SSR markers which are located on different chromosome (CH02c11 on LG10, CH02c06 on LG2). This issue needs to be addressed.

We appreciate the reviewer’s comment. Moreover, the linkage disequilibrium is defined as the ‘occurrence in members of a population of combinations of linked genes in non-random proportions’. It could thus be used, as explained by the reviewer, to evaluate the genetic linkage of the different loci located on the same chromosome. Nevertheless, the ‘inter-chromosomal’ linkage disequilibrium could be also used and the entire manuscript has been revised and the ‘inter-chromosomal’ word has been added to clarify this issue. The three papers below reported (not included in the references section) are examples of different studies with the inter-chromosomal linkage disequilibrium assessed in peach, wheat, and bean, as we have used in our present work in apple:

Cao, K., Wang, L., Zhu, G., Fang, W., Chen, C., & Luo, J. (2012). Genetic diversity, linkage disequilibrium, and association mapping analyses of peach (Prunus persica) landraces in China. Tree Genetics & Genomes, 8, 975-990. DOI: https://doi.org/10.1007/s11295-012-0477-8.

Laido, G., Marone, D., Russo, M.A., Colecchia, S.A., Mastrangelo, A.M., De Vita, P., Papa, R. (2014). Linkage disequilibrium and genome-wide association mapping in tetraploid wheat (Triticum turgidum L.). PloS one, 9(4), e95211. DOI: https://doi.org/10.1371/journal.pone.0095211.

Diniz, A. L., Giordani, W., Costa, Z.P., Margarido, G.R., Perseguini, J.M.K., Benchimol-Reis, L.L., Vieira, M.L.C. (2018). Evidence for strong kinship influence on the extent of linkage disequilibrium in cultivated common beans. Genes, 10(1), 5. DOI: https://doi.org/10.3390/genes10010005.

Table2: Previous QTL analysis indicates that significant QTLs for acidity and polyphenol content are located on LG16, but there is no SSR marker from LG16 in this study. To do association mapping, markers should cover all chromosomes. Please clarify this point.

We appreciate the reviewer’s comment. When working with SSRs we cannot have the same coverage as working with massive genotyping as SNPs. The availability of some SSR sequences is thus sometimes limited.

Nevertheless, we agree with the reviewer’s comment and indeed, in the bibliography there is a QTL or major gene located at LG16 and associated with titratable acidity (TA) and also with ripening index. Unfortunately, we used a SSR (CH04f10) belonging to that LG16 but it did not amplify. Nevertheless, we have found other associations with the TA (LG1, 3, 5, 9, 11, 12, and 14) and the ripening index (LG1, LG5, LG6, LG7, LG9, LG11, LG12, LG14, LG15 and LG17), since these traits are complex characters and their control involves the regulation of many metabolic pathways as acidity. In any case, our results with SSRs would be a good approximation since Urrestarazu et al. (2017) found two NAC transcription factors in the LG3, and also on the LG2 (in the last case close to the CH02c06 SSR marker) as candidate genes for the control of acidity. In our study, as above mentioned, TA was also associated with SSRs at LGs 2 and 3, and RI with SSRs also at LG2, among others). Consequently, we referred the LG16 in our work only as reference or result found by other authors.

Regarding the polyphenol content we can say something similar. We know there are QTLs described in the LG16, but there are some of them also in other groups as reported by Chagné et al. (2012) at LGs 3, 5, and 14, in agreement with those observed in our study.

Thus, we discuss the results found by other authors on LG16, although we have not found any associations in that LG because the SSR CH04f10 belonging to that LG16 did not amplify in our work.

According to the reviewer comment, it is appropriate to note that we have now included this SSR (CH04f109) and other two more SSRs (CH03d07, LG6 and CH01h02, LG9) in Table 2, even though they showed weak or no amplification in our work.

Table5: This result only indicates statistical significance for association in a given marker and a trait. For breeding purpose, we want to know degree of association in terms of phenotypic variation. This reviewer asks to include additional data, which shows relationship of marker genotype and phenotypic variation of a trait.

We appreciate the reviewer’s comment. In this study we have not taken these calculations into account because one of the our near future objective is to continue analysing associations with fine mapping using a large-scale of SNPs markers as genome-wide association studies (GWAS) by the Axiom(®) Apple 480K array. We believe that it would be more convenient to add this information in this publication since it will allow us to validate the results found in the present study with SSRs. In a future study when using SNPs, we will focus on relationship of marker genotype and phenotypic variation of a trait and confirm the LGs of the candidate genes and thus, the results of the present work will take more consistency.

Reviewer 2 Report

The paper titled "Population structure and association mapping for agronomical and biochemical traits of a large Spanish apple germplasm", the authors showed that the phenotypic characters are significant associate with 23 SSR markers.

1. In Figure1, Pectin as one of the most important components of cell wall, did the author detect the pectin cotent in all the accessions phenotyped?

2. Some typical phenotype needs to show picture here.

3. Figure 2, needs high quality.

4. Are these 23 different SSR markers ever report before? Did the authors validate the markers?

Author Response

Comments from Reviewer 2:

  1. In Figure1, Pectin as one of the most important components of cell wall, did the author detect the pectin content in all the accessions phenotyped?

We did not evaluate the pectin content in our accessions. We appreciate the reviewer’s comment but in this study we focused on the determination of the Soluble Solids Content (SSC) expressed as ºBrix, and the determination of total and individual sugars and organic acids using HPLC analysis.

  1. Some typical phenotype needs to show picture here.

We agree with the suggested modification, the comment has been taken into account; and the new ‘Figure 1’ has been added in the Material and Methods section.

Figure 1. Phenotypic fruit diversity encompassed by the 186 accessions of the germplasm bank established at the EEAD-CSIC, Zaragoza, Spain. a: ‘Reineta Gris’; b: ‘Reineta Blanca del Canadá’; c: ‘Granny Smith’; d: ‘Baujade’; e: ‘Morro de Liebre’; f: ‘Golden Paradise’; g: ‘Cripps Pink’; h: ‘Reneta’; i: ‘Solafuente’; j: ‘Evasni’; k: 15 accessions from the EEAD-CSIC germplasm bank exhibiting differences in color and shape.

  1. Figure 2 needs high quality.

We appreciate the reviewer’s suggestion and we have improved the quality of the Figure 2 (currently Figure 3).

Revised Figure 2:

  1. Are these 23 different SSR markers ever report before? Did the authors validate the markers?

We appreciate the reviewer’s comment. Indeed, these 23 SSR have been reported in previous studies (see references section: Pereira-Lorenzo et al., 2017, Urrestarazu et al., 2012, Zhen et al., 2018). The SSRs markers have also been validated in this study demonstrating their discrimination power between accessions and cultivars.

Round 2

Reviewer 1 Report

I appreciate authors for corresponding all the comments I mentioned.

The revised manuscript are now considered acceptable after minor revision. 

P2: Apple is considered to be derived from autopolyploidization not from interspecific hybridization. This sentence needs further revise.

Author Response

Comments from Reviewer 1:

I appreciate authors for corresponding all the comments I mentioned.

The revised manuscript is now considered acceptable after minor revision.

Thank you very much for your valuable help.

P2: Apple is considered to be derived from autopolyploidization not from interspecific hybridization. This sentence needs further revise.

We appreciate this reviewer’s suggestion and agree with his comment. Changes were consequently done and the final sentence in the revised manuscript is: ‘All commercial apple cultivars have a basic chromosome number of 17, and are considered to have evolved after an autopolyploidization of a Gillenia-like taxon followed by diploidization (Gross et al., 2012; Velasco et al., 2010).
